# Genetic Relationships and Signatures of Adaptation to the Climatic Conditions in Populations of *Apis cerana* Based on the Polymorphism of the Gene Vitellogenin

**DOI:** 10.3390/insects13111053

**Published:** 2022-11-15

**Authors:** Rustem A. Ilyasov, Slađan Rašić, Junichi Takahashi, Valery N. Danilenko, Maxim Y. Proshchalykin, Arkady S. Lelej, Vener N. Sattarov, Pham Hong Thai, Rika Raffiudin, Hyung Wook Kwon

**Affiliations:** 1Scientific and Educational Center, Bashkir State Agrarian University, 50-Letiya Oktyabrya Str. 34, 450001 Ufa, Russia; 2Department of Life Sciences, Convergence Research Center for Insect Vectors, Incheon National University, 119 Academy-ro, Yeonsu-gu, Incheon 22012, Republic of Korea; 3Department of Genetics and Biotechnology, Vavilov Institute of General Genetics, Russian Academy of Sciences, Gubkina Str. 3, 119333 Moscow, Russia; 4Faculty of Ecological Agriculture, Educons University, Vojvode Putnika 87, 21208 Sremska Kamenica, Serbia; 5Faculty of Life Sciences, Kyoto Sangyo University, Kamigamo Motoyama, Kita Ward, Kyoto 603-8555, Japan; 6Federal Scientific Center of the East Asia Terrestrial Biodiversity, Prospekt 100-let Vladivostoka, 159, 690022 Vladivostok, Russia; 7Department of Bioecology and Biological Education, Bashkir State Pedagogical University Named after M. Akmulla, 3a October Revolution Street, 450008 Ufa, Russia; 8Research Center for Tropical Bees and Beekeeping, Vietnam National University of Agriculture, Trau Quy, Gia Lam, Hanoi 100000, Vietnam; 9Department of Biology, Bogor Agricultural University, Darmaga, Bogor 16680, Indonesia

**Keywords:** *Apis cerana*, *A. c. koreana*, population, gene vitellogenin *VG*, genetic structure, nucleotide polymorphism, adaptation, Tajima’s D neutrality test

## Abstract

**Simple Summary:**

The Oriental honey bee *Apis cerana,* similar to the Western honey bee *Apis mellifera,* is distributed in different climatic conditions, and each of them is subdivided into more than thirty subspecies and ecotypes. Their sustainability depends on adaptations to the local climate conditions. The *VG* gene is involved in the development, reproduction, labor division, and caste differentiation of honey bees. We found the nucleotide sequences of the *VG* gene reflect the adaptation of honey bees to the local climate conditions. The *VG* gene sequences are acceptable tools to study the sustainability, genetic structure, and adaptation of *A. cerana* populations and can be applied in conservation genetics of local honey bee subspecies.

**Abstract:**

*Apis cerana* and *Apis mellifera* are important honey bee species in Asia. *A. cerana* populations are distributed from a cold, sharply continental climate in the north to a hot, subtropical climate in the south. Due to the Sacbrood virus, almost all *A. cerana* populations in Asia have declined significantly in recent decades and have recovered over the past five years. This could lead to a shift in the gene pool of local *A. cerana* populations that could affect their sustainability and adaptation. It was assumed that adaptation of honey bees could be observed by comparative analysis of the sequences of genes involved in development, labor division, and caste differentiation, such as the gene Vitellogenin *VG*. The *VG* gene nucleotide sequences were used to assess the genetic structure and signatures of adaptation of local populations of *A. cerana* from Korea, Russia, Japan, Nepal, and China. *A. mellifera* samples from India and Poland were used as the outgroup. The signatures of adaptive selection were found in the local population of *A. cerana* using *VG* gene sequence analysis based on Jukes–Cantor genetic distances, cluster analysis, dN/dS ratio evaluation, and Tajima’s D neutrality test. Based on analysis of the *VG* gene sequences, *Apis cerana koreana* subspecies in the Korean Peninsula were subdivided into three groups in accordance with their geographic localization from north to south. The *VG* gene sequences are acceptable tools to study the sustainability and adaptation of *A. cerana* populations.

## 1. Introduction

*Apis cerana* is the second most widely distributed honey bee species in Asia after *Apis mellifera* [1]. *A. cerana* is the basic pollinator and producer of apiculture products in Asia [2]. Similar to *A. mellifera*, *A. cerana* has high potential for further selection based on genetic markers [3]. The native range of *A. cerana* embraces northern to southern Asia from Russia to Indonesia and eastern and western Asia from Japan to Afghanistan [4]. *A. cerana* occupies a large range of climatic conditions at different altitudes and latitudes. Similar to *A. mellifera*, *A. cerana* has high genetic and phenotypic variations across a range and is subdivided into several ecotypes and subspecies [3,5,6]. The most northern distribution of *A. cerana* reaches 47°54′ N latitude in Khabarovsky Krai of Russia [7]. Each subspecies of *A. cerana* in Japan (*A. c. japonica*), in China (*A. c. cerana*), in Korea (*A. c. koreana*), in Russia (*A. c. ussuriensis*), and in Vietnam (*A. c. indica*) are adapted to their native climate and contain genes linked with adaptation [7,8]. There are no published genetic markers connected with adaptation of *A. cerana*. All populations of *A. cerana* are understudied using genetic markers. *A. cerana* subspecies subdivision and their distribution range are studied insufficiently.

The gene pool of *A. cerana* in all populations significantly changed after the Sacbrood virus eliminated 95% of colonies [4,9,10]. Development of agriculture and increased treatment with pesticides have led to additional population declines [3,8,9]. The decline in the populations of *A. cerana* entails a reduction in the biodiversity of local ecosystems. *A. cerana* and *A. mellifera* are not equivalent pollinators of native plants; they cannot replace each other [3,4,11].

Within its natural range, the local subspecies of *A. cerana* has undergone hybridization that violates the adaptive complexes of its gene pool. Additionally, the genetic pool of local populations of subspecies *A. cerana* may be completely replaced by the genetic pool of introduced subspecies *A. cerana*. Conservation of the subspecies *A. cerana* cannot take place without molecular genetic methods. Genetic markers related to adaptation should be found in genes involved in the development and caste differentiation of *A. cerana* [12,13]. Molecular genetic markers make it possible to develop basic strategies for preserving *A. cerana* and study its phylogeography.

Vitellogenin, *VG*, is a basic reproductive protein and endocrine development factor, which, together with juvenile hormone JH and insulin-like growth factor IGF, is responsible for the caste differentiation and development of *A. cerana. VG* is a source of the royal jelly produced by workers and fed to larvae, so just as it affects queen egg laying, it may also affect the rearing of larvae hatching from those eggs. Moreover, *VG* plays an important role in regulating the labor division that can impact colony fitness by affecting colony foraging rates, foraging efficiency, and growth rates. The *VG* gene has high rates of molecular evolution in the honey bee, as evident by high rates of within-*A. mellifera* polymorphisms and high rates of divergence in the genus *Apis*. The *VG* gene is located on the fourth chromosome of *A. mellifera*, which has the highest recombination rate (27.6 cM ⁄ Mb) among all 16 chromosomes. The recombination rate in the *VG* gene is 5.9 times higher than in other genes [12,13,14,15,16]. The experiment with RNAi interference of *VG* gene expression paces feeding behavior and induces shifts in the caste differentiation of *A. mellifera* [17]. The *VG* gene is connected to the spatial distribution and population structure of *A. mellifera* [12,17]. Comparative analysis of *VG* gene showed adaptation signatures in *A. mellifera* populations [13]. The *VG* gene has not been previously studied in *A. cerana* in the aspects of population genetics.

The *VG* gene could be useful for the study of the evolutionary adaptation and population structure of *A. cerana*. Comparative analysis of *VG* gene sequences promises to yield interesting results regarding the relationship and distribution of local *A. cerana* populations in Korea, Russia, Japan, Nepal, and China. The signatures of adaptive selection could be found in the local population of *A. cerana* based on Jukes–Cantor genetic distances, cluster analysis, and Tajima’s D neutrality test. The aim of the paper is to study the genetic signatures of climate adaptation in *Apis cerana* populations using nucleotide sequences of the *VG* gene. The objectives of the paper are to sequence the *VG* gene exons and perform comparative cluster analysis with reference sequences from GenBank.

In this paper, the signatures of adaptive evolution were found in the local population of *A. cerana* using *VG* gene sequence analysis based on Jukes–Cantor genetic distances, cluster analysis, and Tajima’s D neutrality test. The subdivision into three groups in accordance with climate adaptation was found in *Apis cerana koreana* subspecies population in the Korean Peninsula. Thus, the *VG* gene sequences are acceptable tools to study the sustainability and adaptation of *A. cerana* populations.

## 2. Materials and Methods

The worker bees of *A. cerana* were collected from the following five Asian countries: (1) South Korea (provinces Gyeonggido, Gangwondo, Chungcheongbukdo, Gyeongsangbukdo, Jeollabukdo, Gyeongsangnamdo), (2) Japan (prefectures Hiroshima, Hokkaido), (3) Taiwan (Central Taiwan region), (4) Nepal (province Bagmati Pradesh), and (5) Russia (Primorsky Territory) (Figure 1, Appendix A). Worker bees were captured and directly displaced in liquid nitrogen before being stored at −80 °C. The genomic DNA was isolated from the thoraxes of honey bees using the G-spin Total DNA Extraction Kit (iNtRON Biotechnology, South Korea, Cat No. 17045), according to the manufacturer’s instructions at the Division of Life Sciences of Incheon National University. The species of all collected honey bee samples were determined morphologically [5].

The polymerase chain reaction (PCR) with our primers developed for exons of the *VG* gene of *A. cerana* (Table 1) performed in Agilent SureCycler 8800 (Agilent Technologies, Santa Clara, CA, USA) in a 12.5 μL volume included 0.15 μL of TaKaRa Ex-Taq, 1.25 μL of 10 × Ex-Taq buffer, 1 μL of 2.5 mM dNTP mixture, 1 μL of 10 pmol of each primer (Table 1), 0.5 μL of template DNA, and 8.6 μL of nuclease free water under following conditions: 95 °C for 3 min; 35 cycles: 95 °C for 30 s, 60 °C for 30 s, 72 °C for 30 s; and final elongation 72 °C for 10 min.

All PCR products were purified using the QIAquick PCR Purification Kit (250) (QIAGEN, Hilden, Germany) following the instructions of the manufacturer. The PCR products were sequenced on both strands in Macrogen Inc. (Seoul, Korea) using the ABI 3730xl 96-capillary DNA analyzer (Applied Biosystems, Foster City, CA, USA) and the ABI BigDye Terminator v3.1 Cycle Sequencing Kit (Applied Biosystems, Foster City, CA, USA). All sequences were analyzed and submitted to GenBank (National Center for Biotechnology Information, NCBI) (Appendix A). In total, 210 sequences of six exons of 35 honey bee samples (*A. cerana* and *A. mellifera*) were deposited to GenBank (Appendix A).

Comparative analysis of the *VG* gene sequences of *A. cerana* and *A. mellifera* was performed using Clustal W methods of alignment of nucleotide sequences of honey bee samples retrieved from GenBank in MEGA v. 10.0.5 at the Division of Life Sciences of Incheon National University [18]. The sequences of the *VG* gene exons of *A. cerana* and *A. mellifera* were retrieved from GenBank (Appendix A). In order to reduce the number of samples on the dendrogram, the samples from the same regions were combined into regional groups.

Pairwise nucleotide sequence divergences were estimated using Unipro UGENE 1.28 (UNIPRO, Moscow, Russia) and CLC Genomics Workbench 11 (CLCbio, Aarhus, Denmark) based on the *VG* gene sequences with the Jukes–Cantor model [19]. Pairwise Euclidean distances between *A. cerana* populations based on morphology data were calculated using Statistica 8.0. Based on pairwise alignments, amino acid identity (%) was calculated for homologous genes. The significance of Tajima’s D test was determined using coalescent simulations with 10,000 runs as implemented in DNAsp [20].

A phylogenetic tree was constructed using the neighbor-joining method, the Jukes–Cantor model, and 1000 bootstrap replications [21]. The percent of evolutionary divergence, number of nucleotide differences, number of transitions and transversions, and dN/dS ratio of nonsynonymous (dN) to synonymous (dS) substitutions were determined in MEGA X based on the *VG* gene sequences using the Nei–Gojobori method [22].

The dN/dS ratio between all *A. cerana* samples was evaluated. The dN/dS ratio remains one of the most popular and reliable measures of evolutionary pressures in coding regions. The dN/dS ratio provides information about the evolutionary forces operating on a particular gene [23]. The dN/dS ratio quantifies the mode and strength of selection by comparing synonymous substitution rates (dS)—assumed to be neutral—with nonsynonymous substitution rates (dN), which are exposed to selection as they change the amino acid composition of a protein. Under neutrality, dN/dS = 1. For genes that are subject to functional constraint such that nonsynonymous amino acid substitutions are deleterious and purged from the population, i.e., genes under negative selection, dN/dS < 1. For the genes under positive selection, dN/dS > 1 [23].

We performed Tajima’s D neutrality test for *A. cerana* populations from different local populations based on the *VG* gene sequences. Tajima’s D test is a statistical method for testing the neutral mutation hypothesis by DNA polymorphism [24]. Tajima’s D is a statistical method for testing the neutral mutation hypothesis by DNA polymorphism [24]. The purpose of Tajima’s D test is to distinguish between a DNA sequence evolving neutrally without selection and non-neutrally under selection. A neutrally evolving DNA sequence contains mutations with no effect on the fitness and survival of an organism. Tajima’s D test computes a standardized measure of the total number of segregating sites in the sampled DNA and the average number of mutations between pairs in the sample. A negative Tajima’s D signifies an excess of low-frequency polymorphisms relative to expectation, indicating population size expansion after a bottleneck or a selective sweep. A positive Tajima’s D signifies low levels of both low- and high-frequency polymorphisms, indicating a decrease in population size and/or balancing selection [24].

## 3. Results

The *VG* gene sequences of *A. cerana* collected from Korea, Russia, Japan, Nepal, and China were first sequenced (with an average size of 4125 bp) and then aligned with the *VG* gene sequences of *A. cerana* and outgroup sequences of *A. mellifera* (with an average size of 4128bp) retrieved from GenBank (Appendix A, Appendix A). The *VG* gene sequences of *A. mellifera* were used as an outgroup for *A. cerana* samples. Both honey bee species, *A. cerana* and *A. mellifera,* have a similar number and size of *VG* gene exons, and differ only in nucleotide sequences (Appendix A).

All samples of *A. cerana* from Korea, Russia, Japan, Nepal, and China were pooled into twenty populations according to their geographical origin. All three samples of *A. mellifera* from India and Poland were pooled into one outgroup. The grouping of honey bee samples according to their spatial distribution was applied in order to reduce the number of groups so that the phylogenetic tree and tables were not cumbersome and were easy to understand. The population genetic parameters, which included the number of nucleotide differences (transitions + transversions), the ratio of nonsynonymous to synonymous substitutions (dN/dS), and Jukes–Cantor genetic distances, were assessed on twenty-one populations (twenty populations of *A. cerana* and one outgroup population of *A. mellifera*) (Appendix A).

The number of nucleotide differences including transitions + transversions varied from 10 to 61 between *A. cerana* populations and from 249 to 270 between *A. cerana* and *A. mellifera* populations. The minimal number of nucleotide differences including transitions + transversions varied from 10 to 12 between Korean *A. cerana* populations (Goseong–Hongcheon, Goseong–Okcheon, Goseong–Uisung, Gangneung–Samcheok, Samcheok–Sancheong, Samcheok–Okcheon). The maximal number of nucleotide differences including transitions + transversions varied from 50 to 61 between Russian and Chinese (Beijing, Taiwan, Yunnan), Japanese (Kyoto), and Nepalese (Kathmandu) populations of *A. cerana*.

The dN/dS ratio varied from 0.236 to 0.999 between *A. cerana* populations and from 0.496 to 0.538 between *A. cerana* and *A. mellifera* populations. The minimal dN/dS ratio varied from 0.236 to 0.295 between Korean (Samcheok–Goseong, Samcheok–Hongcheon) and Japanese (Samcheok–Hiroshima, Goseong–Hiroshima) *A. cerana* populations. The maximal dN/dS ratio varied from 0.711 to 0.999 between Korean (Hongcheon–Uisung, Samcheok–Wanju), Korean and Japanese (Hongcheon–Kyoto), and Korean and Russian (Samcheok–Vladivostok) *A. cerana* populations.

The Jukes–Cantor genetic distances varied from 0.002 to 0.015 between *A. cerana* populations and from 0.065 to 0.070 between *A. cerana* and *A. mellifera* populations. The minimal Jukes–Cantor genetic distances varied from 0.002 to 0.003 between Korean (Goseong–Hongcheon, Goseong–Samcheok, Goseong–Sancheong, Goseong–Okcheon, Goseong–Uisung, Gangneung–Samcheok, Hongcheon–Samcheok, Hongcheon–Uisung, Samcheok–Okcheon, Gangneung–Sancheong, Hongcheon–Okcheon, Gangneung–Wanju, Sancheong–Wanju, Seoul–Wanju) *A. cerana* populations. The maximal Jukes–Cantor genetic distances varied from 0.012 to 0.015 between Russian and Korean (Vladivostok–Uisung, Vladivostok–Goseong), Russian and Japanese (Vladivostok–Kyoto), Russian and Chinese (Vladivostok–Jiangxi, Vladivostok–Yunnan), and Russian and Nepalese (Vladivostok–Kathmandu) *A. cerana* populations.

The phylogenetic tree was constructed using *VG* gene sequences based on Jukes–Cantor genetic distances using the neighbor-joining method of clustering and 1000 bootstrap replications (Figure 2). Korean *A. cerana* populations were subdivided into three clusters. Japanese *A. cerana* populations were distantly joined to the Korean *A. cerana* group. All Chinese and Nepalese *A. cerana* populations were clustered together. The Russian *A. cerana* population was located apart from all *A. cerana* populations (Figure 2).

In addition, the forty samples of *A. cerana* from Korea, Russia, Japan, Nepal, and China were grouped into five major populations according to the location of the country. All three samples of *A. mellifera* from India and Poland were pooled into one outgroup as representatives of one sister species. The aim of the paper is to study populations of *A. cerana* in comparison with outgroup sister species *A. mellifera*. The grouping of *A. cerana* samples into five major populations and *A. mellifera* samples into one major outgroup population provides insight into the global population structure of *A. cerana* in Asia. The population genetic parameters, which included the number of nucleotide differences (transitions plus transversions), the ratio of nonsynonymous to synonymous substitutions (dN/dS), and Jukes–Cantor genetic distances, were assessed on six populations (five populations of *A. cerana* and one outgroup population of *A. mellifera*) (Table 2).

The average number of nucleotide differences, including transitions and transversions, varied from 20.6 to 54.3 between *A. cerana* populations and from 255.1 to 269.8 between *A. cerana* and *A. mellifera* populations. The minimal number of nucleotide differences, including transitions and transversions, varied from 20.6 to 28.8 between Japanese–Korean and Japanese–Nepalese *A. cerana* populations. The maximal number of nucleotide differences, including transitions and transversions, varied from 51.5 to 54.3 between Russian–Chinese and Russian–Nepalese *A. cerana* populations (Table 2).

The average dN/dS ratio varied from 0.236 to 0.999 between *A. cerana* populations and from 0.496 to 0.538 between *A. cerana* and *A. mellifera* populations. The minimal dN/dS ratio varied from 0.236 to 0.295 between Korean (Samcheok–Goseong, Samcheok–Hongcheon) and Japanese (Samcheok–Hiroshima, Goseong–Hiroshima) *A. cerana* populations. The maximal dN/dS ratio varied from 0.711 to 0.999 between Korean (Hongcheon–Uisung, Samcheok–Wanju), Korean and Japanese (Hongcheon–Kyoto), and Korean and Russian (Samcheok–Vladivostok) *A. cerana* populations.

The average number of nucleotide differences based on the *VG* gene sequences, the dN/dS ratio between all *A. cerana* and the *A. mellifera* outgroup populations, varied from 20.6 to 269.8 (Table 2 and Appendix A). The number of nucleotide differences between *A. cerana* populations from different countries varied from 20.6 to 28.8. The maximal number of nucleotide differences varied from 255.1 to 269.8 between *A. cerana* and *A. mellifera* outgroup populations. The minimal number of nucleotide differences was 20.6 between Korean and Japanese populations of *A. cerana* (Table 2 and Appendix A).

The average Jukes–Cantor genetic distances between local populations of *A. cerana* and *A. mellifera* varied from 0.005 to 0.070 (Table 2 and Appendix A). The genetic distances between *A. cerana* populations varied from 0.005 to 0.014. The maximal genetic distances varied from 0.066 to 0.070 and were observed between *A. cerana* and *A. mellifera* outgroup populations (Table 2 and Appendix A). The minimal genetic distance was 0.005 between the Korean and Japanese populations of *A. cerana*. The maximal genetic distances varied from 0.010 to 0.014 between the Chinese, Nepalese, and Russian populations of *A. cerana* (Table 2 and Appendix A).

We constructed the neighbor-joining phylogenetic tree and the three-dimensional PCA plot based on Jukes–Cantor genetic distances between *VG* gene sequences of five *A. cerana* and one outgroup *A. mellifera* populations (Figure 3A,B). *A. cerana* populations from Korea and Japan were clustered together and separately from populations from Russia, Nepal, and China (Figure 3A).

The Tajima’s neutrality test was assessed for all *A. cerana* populations based on the *VG* gene sequences (Table 3). Additionally, the number of segregating sites, nucleotide diversity, number of synonymous and nonsynonymous sites, and the number of nucleotide differences were counted on the basis of the *VG* gene sequences (Table 3).

## 4. Discussion

The genetic structure of the *A. cerana* populations in the Korean Peninsula was insufficiently studied previously using nuclear genetic markers. The Korean Peninsula is a predominantly mountainous landscape with various altitudes, ranging from a continental climate in the north to a subtropical climate in the south. South Korea, part of the East Asian Monsoon region, has a temperate climate with four distinct seasons. The movement of air masses from the Asian continent exerts a greater influence on South Korea’s weather than air movement from the Pacific Ocean. Winters are usually long, cold, and dry, whereas summers are short, hot, and humid. Spring and autumn are rather short [25].

In this project, we analyzed *A. cerana* from a wide range of the Korean Peninsula, from south to north and from east to west in a comparison with *A. cerana* from northern (Russia, Japan) and southern (China, Nepal) countries and outgroup samples of *A. mellifera*, to find genetic signatures of adaptations to the local environments based on the nucleotide polymorphism of the nuclear *VG* gene. Both *A. cerana* and *A. mellifera* species have similar exon–intron structures and differ only by nucleotide sequences (Figure 2). This makes it possible to provide comparative analysis samples from different regions with adaptation to cold and hot climates. Due to the *VG* gene’s involvement in development and behavior, its sequences can reflect local adaptations to various climates. Samples of *A. cerana* adapted to similar environmental conditions should have fewer genetic differences in *VG* gene sequences than samples adapted to different environmental conditions. According to the *VG* gene analysis, populations of *A. cerana* from northern Asian countries, such as China, Japan, Korea, and Russia, grouped together and separately from populations of *A. cerana* from southern Asian countries, such as Nepal.

The genetic analysis was provided with 40 samples of *A. cerana* from Korea, Russia, Japan, Nepal, and China from 20 geographical populations and outgroup *A. mellifera* samples from India and Poland. The average number of nucleotide differences between the *VG* gene sequences, including transitions and transversions, varied from 20.6 to 28.8 between *A. cerana* populations from different countries, which is an indication of the genetic divergence level. The minimal number of nucleotide differences (20.6) was observed between Korean and Japanese populations of *A. cerana* (Appendix A), which proves their close genetic relationship due to recent active movement of bee colonies between countries. The nucleotide differences between the *A. cerana* and *A. mellifera* outgroup populations were highest (255.1–269.8), which demonstrates their significant divergence. We counted the number of nucleotide differences between the *VG* gene sequences within the northern populations of *A. cerana*, which varied from 10 to 52 (the average was 21.9). Additionally, we counted the number of nucleotide differences in the *VG* gene sequences between the northern and southern populations of *A. cerana*, which varied from 22 to 61 (the average was 32.4) (Appendix A). Here, the majority of the nucleotide differences of the *VG* gene sequences between the northern and southern populations of *A. cerana* were higher than those within the northern populations of *A. cerana*. The number of nucleotide substitutions likely reflects the magnitude of local adaptations, which are minimal between populations from similar climatic conditions. Additionally, the similarity in *VG* gene sequences can be caused by human-assisted movement of *A. cerana* colonies between countries.

The dN/dS ratio between all *A. cerana* samples varied from 0.158 to 0.999, which demonstrates the functional significance of the *VG* gene such that nonsynonymous amino acid substitutions are deleterious and purged from the population. Indeed, the *VG* gene affects multiple aspects of social organization of honey bee colonies and plays a key role in the caste differentiation and maintenance of honey bees. The dN/dS ratio between the northern populations of *A. cerana* varied from 0.158 to 0.999 (the average was 0.46), which indicates strong negative selection for the *VG* gene, which characterizes the high significance of the *VG* gene’s conservativeness in the adaptation of *A. cerana* to the cold northern climate. Positive selection leads to fixation changes in genes, and negative, or purifying, selection leads to rejection changes in genes. The dN/dS ratio between the northern and southern populations of *A. cerana* varied from 0.289 to 0.947 (the average was 0.49) (Appendix A), which indicates less strong negative selection of the *VG* gene in evolution and the high significance of the *VG* gene polymorphism in the adaptation of *A. cerana* to the different climates of southern and northern Asia. In comparison, the average dN/dS ratio for 3256 nuclear genes was 0.1068 between 3 species of Formicidae, 0.1033 between 3 species of Polistinae and Vespinae wasps, and 0.1086 between 10 species of *Apis*, *Bombus*, and *Tetragonula*. It was 0.1025 between three species of Siricoidea and 0.1005 between five species of Cynipoidea, which characterized the negative selection of these genes in evolution and their role in adaptation to the local climate [26]. It has been reported that genes involved in the labor division, caste differentiation, and development are affected by strong negative selection in social insects. Authors estimated the strength of negative selection at 5′ UTRs, 3′ UTRs, introns, exons, intergenic sites, and overall genomes in *A. mellifera*, *B. impatiens*, *P. dominula*, and *S. invicta* and found that 0-fold sites with nonsynonymous changes and overall genomes experienced the strongest negative selection in all four eusocial species [27]. On the contrary, another study showed evidence for positive selection on the *VG* gene in *A. mellifera* and *A. cerana*, which leads to fixation of the nonsynonymous mutations, bias of the allele frequency spectrum, growth of the genetic differentiation at neutral sites, and magnification of the linkage disequilibrium relative to other genes [13]. Hence, the *VG* gene is very important for the adaptation of *A. cerana*; most nucleotide variations reduce its sustainability and are eliminated from populations.

The Jukes–Cantor genetic distance is the genetic relatedness between two nucleotide sequences calculated as the sum of the substitutions that have accumulated between them since they diverged from their common ancestor during evolution, assessed using the Jukes–Cantor model. The Jukes–Cantor genetic distances based on the *VG* gene sequences between *A. cerana* populations varied from 0.002 to 0.015, and between *A. cerana* and *A. mellifera* populations varied from 0.065 to 0.070. The Jukes–Cantor genetic distances based on the *VG* gene sequences between the northern populations of *A. cerana* varied from 0.002 to 0.013 (the average was 0.005). The Jukes–Cantor genetic distances based on the *VG* gene sequences between the northern and southern populations of *A. cerana* varied from 0.005 to 0.015 (the average was 0.010) (Appendix A). In comparison, the Jukes–Cantor genetic distance was 0.018 between subspecies of *A. mellifera* (*A. m. carnica*, *A. m. ligustica*, *A. m. sicula*, *A. m. iberica*, *A. m. adami*, *A. m. macedonica*, *A. m. anatoliaca*, *A. m. syriaca*, *A. m. intermissa*) based on the *COX1* gene of mtDNA [28]; was 0.17 between 2 honey bee species (*A. mellifera*, *A. cerana*) based on complete mtDNA [29]; varied from 0.001 to 0.014 between subspecies of *A. cerana*; varied from 0.075 to 0.081 between species *A. mellifera* and *A. cerana* based on the *VG* gene of nDNA [8]; was 0.033 between 15 species of *Anastrepha* based on the *COX1* gene of mtDNA [30]; and varied from 0.002 to 0.039 within species and from 0.074 to 0.131 between 5 species of *Apis* based on the *CYTB* gene of mtDNA [31]. Thus, the Jukes–Cantor genetic distances based on the *VG* gene sequences between *A. cerana* populations from similar climatic conditions is half that between *A. cerana* populations from different climatic conditions. This can explain the important role of the *VG* gene in the adaptation of *A. cerana* populations to different climatic conditions from northern to southern Asia. The phylogenetic tree constructed using the neighbor-joining method with the Jukes–Cantor genetic distances based on the *VG* gene sequences (Figure 3) shows the informativeness of the gene in phylogenetic and ecological studies of *A. cerana*. Additionally, we can see the genetic relationship between *A. cerana* populations on the phylogenetic tree and three-dimensional PCA plot (Figure 3). Moreover, there was a genetic differentiation based on the *VG* gene sequences between Korean populations of *A. c. koreana* that is related to the geographic localization and climatic conditions of southern, central, and northern provinces of the Republic of Korea.

The results of Tajima’s D test were counted for each population of *A. cerana*. Almost all of the estimated D values in *A. cerana* bees were not significantly different from zero, implying that *A. cerana* honey bees are in a state of mutation drift equilibrium [17]. The number of segregating sites (S) was lowest in the Nepalese population of *A. cerana* and highest in the Chinese population of *A. cerana*. This may be due to the high genetic diversity that should be observed in the largest Chinese population of *A. cerana*. The nucleotide diversity was highest in the Russian population of *A. cerana* and lowest in the Korean population of *A. cerana*. This can be explained by the fact that the *A. cerana* population in Russia is feral under natural selection. There are no managed *A. cerana* populations in Russia (Table 3).

The values of Tajima’s D test in all *A. cerana* populations were statistically significant. The Korean population of *A. cerana* had the biggest negative value (−0.685), which means that this population could be in a state of bottleneck, selective sweep, purifying selection, and negative selection. Indeed, the Korean population of *A. c. cerana* experienced an extreme population decline with the kSBV virus, which killed 95% of the local population [4,9,10,32]. All other populations of *A. cerana* also had the signatures of the population size expansion after a selective sweep in the past (Table 3). For comparison, Tajima’s D test values based on mitochondrial *COX1* gene sequences varied from −1.369 to 2.383 in the Guangdong Province population of *A. c. cerana*, from −2.293 to −0.048 in the Guangxi Autonomous Region population of *A. c. cerana*, from −1.670 to 1.276 in the Hainan Island population of *A. c. cerana*, and from −1.622 to 2.383 in the mainland China population of *A. c. cerana* [33]. The values of Tajima’s D test in Russian, Korean, Japanese, Chinese, and Nepalese populations of *A. cerana* based on *VG* gene sequences were similar to those in China based on the mitochondrial *COX1* gene. Thus, both nuclear and mitochondrial genes showed similar results for *A. cerana* populations, which were characterized by an excess of low-frequency polymorphisms relative to expectations, indicating population size expansion after a bottleneck due to the Sacbrood virus. The Tajima’s D test values and dN/dS ratio can be used as indicators for negative selection and the role of some genes in adaptation of local *A. cerana* populations [33].

## 5. Conclusions

We performed a thorough genetic analysis on *A. cerana* populations from South Korea in comparison with *A. cerana* populations from Russia, Japan, Nepal, and China and the outgroup population of *A. mellifera*. Based on the *VG* gene sequences, we assessed the negative selection that characterizes adaptation in *A. cerana* populations to the local climatic conditions. The cluster and principal component analysis based on the sequences of the *VG* gene divided populations of *A. cerana* according to their climatic and geographic distribution into southern and northern groups. The local populations of *A. c. koreana* were subdivided according to their geographical distribution into southern, northern, and central Korean clusters. The Jukes–Cantor genetic distances based on the *VG* gene sequences showed the *A. cerana* populations from northern countries such as Korea, Russia, and Japan are more closely related to each other than the southern populations of *A. cerana*. The signatures of adaptation were found in the local population of *A. cerana*. All *A. cerana* populations showed signs of population size expansion following recent declines. The *VG* gene sequences can be used as informative markers for monitoring the genetic structure and adaptation processes in *A. cerana* populations.

## Figures and Tables

**Figure 1 insects-13-01053-f001:**
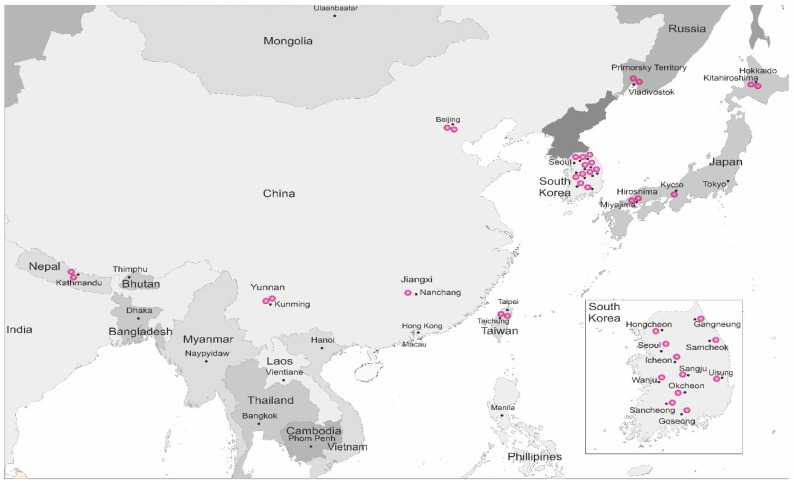
Spatial distribution of collected *A. cerana* samples in Asia.

**Figure 2 insects-13-01053-f002:**
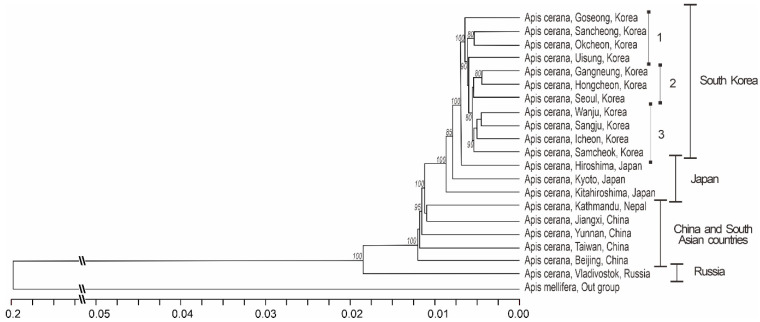
A neighbor-joining phylogenetic tree of *A. cerana* samples with outgroup *A. mellifera* based on the *VG* gene sequences. The Korean *A. cerana* populations subdivided into three groups according to their geographical distribution: 1. the southern provinces; 2. the northern provinces; and 3. the central provinces.

**Figure 3 insects-13-01053-f003:**
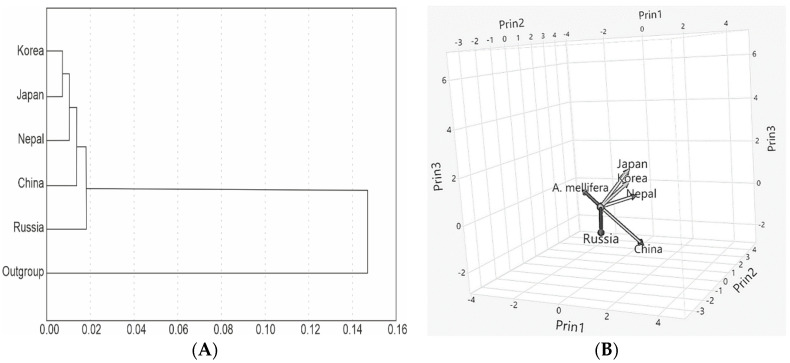
Neighbor-joining phylogenetic tree (**A**) and three-dimensional PCA plot (**B**) based on Jukes–Cantor genetic distances between *VG* gene sequences of five *A. cerana* and one outgroup *A. mellifera* populations.

**Table 1 insects-13-01053-t001:** PCR primers for the *VG* gene exons of *Apis cerana*.

Name	Sequence 5′-3′	TM, °C	GC, %	Size, bp
Ac*VG*E2-F	TCTTGTTCGTTCCAGGTTCC	58.4	50	677
Ac*VG*E2-R	GACAGTTTCAGCCGACTTCC	60.5	55	
Ac*VG*E3-F	CCTTTCGATCCATTCCTTGA	56.4	45	679
Ac*VG*E3-R	GTCAAAACGGATTGGTGCTT	56.4	45	
Ac*VG*E4-F	TCGAAGGGGAAGAATTTCAA	54.3	40	840
Ac*VG*E4-R	ACGAGCAATTCCTCAACACC	58.4	50	
Ac*VG*E5-F	GTCGGACAATTTCACGTCCT	58.4	50	1177
Ac*VG*E5-R	GTTCGAGCATCGACACTTCA	58.4	50	
Ac*VG*E6-F	AGAGCCAGGGATACGTCAAA	58.4	50	406
Ac*VG*E6-R	GAGTCATCTCGAGGCTCACC	62.5	60	
Ac*VG*E7-F	TTCTGGCTGAGGTCAGGATT	58.4	50	445
Ac*VG*E7-R	AATTTCGACCACGACTCGAC	58.4	50	

**Table 2 insects-13-01053-t002:** The Jukes–Cantor genetic distances above the diagonal, the number of nucleotide differences (transitions + transversions), and the ratio of nonsynonymous to synonymous substitutions (dN/dS) below the diagonal between populations of *A. cerana* and outgroup *A. mellifera* based on the *VG* gene sequences and their standard error S.E.

Populations		Korea	China	Japan	Nepal	Russia	*A. mellifera* Outgroup
	Jukes Cantor Genetic Distances/S.E.						
Korea	Number of nucleotide differences (tr + tv)/S.E. [dN/dS ratio/S.E.]		0.008/0.001	0.005/0.001	0.007/0.001	0.010/0.001	0.066/0.004
China		31.5/3.8 (0.49/0.001)		0.009/0.001	0.008/0.001	0.014/0.002	0.066/0.004
Japan		20.6/3.1 (0.39/0.001)	34.0/4.1 (0.42/0.001)		0.007/0.001	0.011/0.001	0.067/0.004
Nepal		28.8/4.7 (0.52/0.001)	31.0/4.2 (0.76/0.001)	28.8/4.8 (0.50/0.001)		0.013/0.002	0.068/0.004
Russia		40.7/5.4 (0.68/0.002)	54.3/5.35 (0.58/0.002)	42.2/5.6 (0.63/0.002)	51.5/6.7 (0.70/0.002)		0.070/0.005
*A. mellifera* outgroup		255.9/5.4 (0.52/0.005)	255.1/5.3 (0.51/0.004)	256.7/5.6 (0.51/0.005)	261.3/6.0 (0.52/0.005)	269.8/6.6 (0.50/0.005)	

Notes: S.E.—standard error. *p* ≤ 0.05.

**Table 3 insects-13-01053-t003:** Tajima’s neutrality test for five *A. cerana* populations based on the *VG* gene sequences.

Population	S	Ps	Θ	π	D	dN	dS	dN/dS	d	S.E.
Korea	53	0.013	0.003	0.004	−0.685	0.003	0.007	0.448	16.692	0.003
Russia	61	0.015	0.010	0.010	−0.023	0.002	0.005	0.440	12.000	0.003
China	80	0.019	0.008	0.008	−0.252	0.006	0.013	0.462	36.667	0.003
Japan	46	0.011	0.005	0.005	−0.041	0.004	0.009	0.408	22.200	0.001
Nepal	32	0.008	0.005	0.005	−0.042	0.003	0.002	1.263	10.000	0.004

Notes: S—number of segregating sites, Ps—S/n, Θ—Ps/a1, π—nucleotide diversity, D—the Tajima test statistic, dS—synonymous sites, dN—nonsynonymous sites, d—the number of nucleotide differences, S.E.—standard error. *p* ≤ 0.05.

## Data Availability

All *A. cerana* and *A. mellifera VG* gene sequences are available in the GenBank (accession numbers MH755714-MH755923).

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
