# Peer review of "Genetic Relationships and Signatures of Adaptation to the Climatic Conditions in Populations of Apis cerana Based on the Polymorphism of the Gene Vitellogenin"

_insects, 2022, doi:10.3390/insects13111053_

Round 1

Reviewer 1 Report

First of all, I would like to thank the authors for letting me read their work and congratulate them for the work that they have already put in this piece. I ask the authors to have into account that my remarks are advices not to put  down their research, but to try to help them improve their work and show more clearly the fruits of their labor, with the final objective of having clear and consistent data and conclusions to further expand their field and questions.

Overall regarding the form of the article, one of the major comment I have to give is that the text could become more fluid and easier to read by changing the use of punctuation marks and connectors to rephrase sentences through the text. An English revision is necessary, as although the main problem is the construction of some paragraphs and ideas, there are some incorrections that need to be changed. The introduction, material and methods are incomplete, with some parts of the results and discussion in need to be rearranged, and the objectives and hypothesis missing. Moreover, the discussion needs to be rewritten and made clearer.

I advise the authors to rethink the structure of the text, remembering which are the key points of each of the parts (introduction: context, importance and objectives; material and methods: justification of techniques and protocols in order to be replicated if necessary; results: presentation of the results; and discussion: scientific context of the results and resolution of questions asked in the introduction).

About the quality of the data, the differences between the groups are presented solidly. I will leave to the rest of reviewers to decide if the sample sizes are consistent to publication, as from my background in population genetics would be insufficient, but my expertise does not reside in working with nucleotide sequences.

Please find here my list of comments:

-Line 27 Reprhase: “A. cerana populations have sharply declined in all Asian countries as a result of Sacbrood Virus infection and have now recovered to their original size. It can change the genetic structure of local populations of A. cerana.”

-Line 50: The phrase about “the genetic improvement of the bee popualtions” comes without enough context. I recommend first, describing the species, ecology, distribution and background about their genetic structure before discussing a clear application of the data. In case that this aim to “improve” a species is one of the reasons to further study the topic, it should be brought when discussing the definition of objectives and the questions that this paper aims to answer.

 I also recommend, given the broad audience of the journal, to use carefully words like “genetic improvement” and, if used, this being done only with proper context and explanation; as we are discussing a clear commercial perspective: from a evolutive or conservation perspective a species genetic pool cannot be “improved”.

-Line 65 to line 73: This is a key paragraph in the introduction, the threats and drivers of decline. The structure needs to be improved. The text starts enlisting some threats, to pass to a broader perspective “Within its native range, A. cerana requires conservation efforts.” and then continues listing threats. Please reorder and add the reasons why these drivers are threats for the species (for example, why is kSBV a problem? You can discuss it by informing of its effects or what impact it currently has on populations). A good perspective to have into account in the introduction is to go from broad problems to the specifics.

-Line 74: What are the objectives and hypotheses of this article? In this article it is discussed why the methodology was selected, but not what are the reasons to do the research.

-Line 91: typo in -800 degrees Celsius (-80ºC).

-Line 96 and Table1: please add the reference from where your selected primers belong or if they have been designed or modified by the authors.

-Line 107: to Genbank

-Line 112: the use of the construct “the gene Vitellogenin exons” is confusing. As you have already explained the acronym before, you can use “VG sequences” or “VG exon sequences”.

-Line 112: was/were, not has been (depending on if analysis is singular or plural).

-Line 115: “In total, 48 sequences of six exons of eight honey bee samples (A. cerana and A. mellifera) were retrieved from GenBank” is slightly confusing, p lease rephrase. And is also problematic, as the sample size is later discussed to be 40 (35+8=43).

-Line 120 and Line 121: Authors discuss their use of “complete mtDNA sequences” and “morphologic data” but they have not previously discussed in the introduction or material and methods from where this data comes from and how or why is being used (Does it belong to the same samples? Does it come from genebank or other databases?). I fail to recall when it is used or for what analysis.

-Line 126-Line 131: Please rephrase.

-Line 135: samples were first sequenced (with an average size of 4125bp) and then aligned. “with retrieved from genbank sequences of seven samples” is not a correct structure.

-Line 143: when alleged that samples have been grouped based on geographical origin does that mean country? Or location? Please define and discuss what the grouping is based on. This information belongs to material and methods, not to the results. Which analyses were affected by this grouping?

-Line 143-150: If possible, I would like to know why the authors have grouped the samples, and on the same note what was the diversity found between groups.

-Line 143: It would improve the understanding of the article and supplementary material to add a table with population coding, population size, location and species. If authors find it interesting, they can also add an asterisk to those populations with information partially/totally retrieved from genbank. Moreover, if space is a problem, I recommend adding Table 1 to the supplementary material.

-I recommend the use of PopArt or a similar software to add an haplotype network, which would enhance the resolution of the results and geographical differences between the haplotypes. If sample sizes were bigger I would recommend studying nucleotide divergence in search for significative differences as depicted for example in Blasco et al. (2019).

-Line 191: Did all the outgroup samples showed the same sequence? In Figure 2 we can only see one entry for “A. mellifera outgroup”, as well as the 20 populations instead of the 35 samples. Does this mean that the región was completely identical between sampling groups? That would be an interesting remark to do and could partially justify the sample sizes. If that is not the case, it needs to be discussed why the rest of the samples were not added.

-Line 194: For this analysis, the samples have been grouped again, now based on the country. First of all, information about the grouping of samples should be discussed in Material and Methods. Second, it makes it more difficult to understand the results. I recommend the authors to discuss one grouping of the samples and continue its use along the whole article, unless there are clear reasons to do so, or limitations to the software in use.

Blasco-Lavilla, N., Ornosa, C., Michez, D. et al. Contrasting patterns of genetic and morphological diversity in the bumblebee Bombus lucorum (Hymenoptera: Apidae: Bombus) along a European gradient. J Insect Conserv 23, 933–943 (2019). https://doi.org/10.1007/s10841-019-00178-2

-Line 249: This paragraph belongs to the end of the introduction, as you discuss the aim and importance of the research.

-Line 281: “pleasant” is not an adequate word to scientifically describe weather.

-Discussion until line 287: you are describing the environment of the species, this should have been done before (if it is truly necessary, as is not discussed later and does not give important insight).

-Line 288: “…wide range in Korea” is enough to describe the sampling.

-Line 297: this is one of your hypotheses, it should be in the introduction.

-Line 322, 383: This belongs in material and methods as why you have selected the technique and should be done for the rest of the techniques.

-Line 343: This is a good way to show the importance of your data, the comparison with other taxa makes clearer the differences found in this article. I would recommend the authors to expand on this perspective, discussing their results with other published work in the region, as well as the differences found between A. mellifera populations and other taxa as explained in line 359.

Author Response

Reply to the Reviewer 1

First of all, I would like to thank the authors for letting me read their work and congratulate them for the work that they have already put in this piece. I ask the authors to have into account that my remarks are advices not to put  down their research, but to try to help them improve their work and show more clearly the fruits of their labor, with the final objective of having clear and consistent data and conclusions to further expand their field and questions.
- Thanks to the first reviewer for carefully reading our manuscript. Thank you very much for helping in improving our manuscript. We appreciate your work. All corrections are green highlighted.

Overall regarding the form of the article, one of the major comment I have to give is that the text could become more fluid and easier to read by changing the use of punctuation marks and connectors to rephrase sentences through the text. An English revision is necessary, as although the main problem is the construction of some paragraphs and ideas, there are some incorrections that need to be changed. The introduction, material and methods are incomplete, with some parts of the results and discussion in need to be rearranged, and the objectives and hypothesis missing. Moreover, the discussion needs to be rewritten and made clearer.
- Okay, thanks, you are right. We changed English text to more fluid and easier to read.

I advise the authors to rethink the structure of the text, remembering which are the key points of each of the parts (introduction: context, importance and objectives; material and methods: justification of techniques and protocols in order to be replicated if necessary; results: presentation of the results; and discussion: scientific context of the results and resolution of questions asked in the introduction).
- Thank you, okay, we revised the structure of the text.

About the quality of the data, the differences between the groups are presented solidly. I will leave to the rest of reviewers to decide if the sample sizes are consistent to publication, as from my background in population genetics would be insufficient, but my expertise does not reside in working with nucleotide sequences.
- Thank you for the suggestion. Usually in population genetics with nucleotide sequences such sample sizes are sufficient.

I also recommend, given the broad audience of the journal, to use carefully words like “genetic improvement” and, if used, this being done only with proper context and explanation; as we are discussing a clear commercial perspective: from a evolutive or conservation perspective a species genetic pool cannot be “improved”.
- Thank you for the recommendation. We agree. The words “genetic improvement” we will use carefully. We revised text.

Line 27 Reprhase: “A. cerana populations have sharply declined in all Asian countries as a result of Sacbrood Virus infection and have now recovered to their original size. It can change the genetic structure of local populations of A. cerana.”
-the sentences are rephrased.

Line 50: The phrase about “the genetic improvement of the bee popualtions” comes without enough context. I recommend first, describing the species, ecology, distribution and background about their genetic structure before discussing a clear application of the data. In case that this aim to “improve” a species is one of the reasons to further study the topic, it should be brought when discussing the definition of objectives and the questions that this paper aims to answer.
-The phrases about the genetic improvement are revised.

Line 65 to line 73: This is a key paragraph in the introduction, the threats and drivers of decline. The structure needs to be improved. The text starts enlisting some threats, to pass to a broader perspective “Within its native range, A. cerana requires conservation efforts.” and then continues listing threats. Please reorder and add the reasons why these drivers are threats for the species (for example, why is kSBV a problem? You can discuss it by informing of its effects or what impact it currently has on populations). A good perspective to have into account in the introduction is to go from broad problems to the specifics.
-the structure is improved, introduction is revised.

Line 74: What are the objectives and hypotheses of this article? In this article it is discussed why the methodology was selected, but not what are the reasons to do the research.
-the objectives and hypotheses are added.

Line 91: typo in -800 degrees Celsius (-80ºC).
-it is corrected.

Line 96 and Table1: please add the reference from where your selected primers belong or if they have been designed or modified by the authors.
-primers are developed in the Sensory Neurobiology and Biomodeling Laboratory in Korea.

Line 107: to Genbank
-it is corrected.

Line 112: the use of the construct “the gene Vitellogenin exons” is confusing. As you have already explained the acronym before, you can use “VG sequences” or “VG exon sequences”.
-it is corrected.

Line 112: was/were, not has been (depending on if analysis is singular or plural).
-it is corrected.

Line 115: “In total, 48 sequences of six exons of eight honey bee samples (A. cerana and A. mellifera) were retrieved from GenBank” is slightly confusing, p lease rephrase. And is also problematic, as the sample size is later discussed to be 40 (35+8=43).
-it is corrected.

Line 120 and Line 121: Authors discuss their use of “complete mtDNA sequences” and “morphologic data” but they have not previously discussed in the introduction or material and methods from where this data comes from and how or why is being used (Does it belong to the same samples? Does it come from genebank or other databases?). I fail to recall when it is used or for what analysis.
-mtDNA here was written erroneously. it is corrected.

Line 126-Line 131: Please rephrase.
-it is rephrased.

Line 135: samples were first sequenced (with an average size of 4125bp) and then aligned. “with retrieved from genbank sequences of seven samples” is not a correct structure.
-it is corrected.

Line 143: when alleged that samples have been grouped based on geographical origin does that mean country? Or location? Please define and discuss what the grouping is based on. This information belongs to material and methods, not to the results. Which analyses were affected by this grouping?
-the explanation on grouping is added. Also added to the material and methods.

Line 143-150: If possible, I would like to know why the authors have grouped the samples, and on the same note what was the diversity found between groups.
-the explanation on grouping is added. There is diversity between groups.

Line 143: It would improve the understanding of the article and supplementary material to add a table with population coding, population size, location and species. If authors find it interesting, they can also add an asterisk to those populations with information partially/totally retrieved from genbank. Moreover, if space is a problem, I recommend adding Table 1 to the supplementary material.
-the article was improved. There is no problem with space, Table 1 is not replaced into supplementary material.

I recommend the use of PopArt or a similar software to add an haplotype network, which would enhance the resolution of the results and geographical differences between the haplotypes. If sample sizes were bigger I would recommend studying nucleotide divergence in search for significative differences as depicted for example in Blasco et al. (2019).
-the manuscript used cluster analysis, which allows clearly separate and observe groups.

Line 191: Did all the outgroup samples showed the same sequence? In Figure 2 we can only see one entry for “A. mellifera outgroup”, as well as the 20 populations instead of the 35 samples. Does this mean that the región was completely identical between sampling groups? That would be an interesting remark to do and could partially justify the sample sizes. If that is not the case, it needs to be discussed why the rest of the samples were not added.
-the aim of our paper is study Apis cerana. Apis mellifera was used only as outgroup. If we show all samples without grouping it will complicate the understanding the figures.

Line 194: For this analysis, the samples have been grouped again, now based on the country. First of all, information about the grouping of samples should be discussed in Material and Methods. Second, it makes it more difficult to understand the results. I recommend the authors to discuss one grouping of the samples and continue its use along the whole article, unless there are clear reasons to do so, or limitations to the software in use.
-we discussed about all grouping reasons.

Line 249: This paragraph belongs to the end of the introduction, as you discuss the aim and importance of the research.
-this paragraph added to the introduction.

Line 281: “pleasant” is not an adequate word to scientifically describe weather.
-this word is replaced.

Discussion until line 287: you are describing the environment of the species, this should have been done before (if it is truly necessary, as is not discussed later and does not give important insight).
-we removed redundant information.

Line 288: “…wide range in Korea” is enough to describe the sampling.
-it is corrected.

Line 297: this is one of your hypotheses, it should be in the introduction.
-it is added to the introduction.

Line 322, 383: This belongs in material and methods as why you have selected the technique and should be done for the rest of the techniques.
-it is added to the material and methods.

Line 343: This is a good way to show the importance of your data, the comparison with other taxa makes clearer the differences found in this article. I would recommend the authors to expand on this perspective, discussing their results with other published work in the region, as well as the differences found between A. mellifera populations and other taxa as explained in line 359.
-we added additional information into discussion.

Reviewer 2 Report

The authors examined the exons of honeybee Vg gene. This gene is very important in honey bees and showed quite some diversity. This study examines the variation of Vg genes in A. cerana populations, with an emphesis on those from northern distribution of A. cerana.

I do have some concers over some of the expressions in the manuscript where at least overstate of importance is found:

(1) line 61-64: the phrase "A. c. koreana is the most interesting ...", "most of the honeybee migrations take place through the Korean peninsula"

(2) line 250: "all beekeepers and scientists are interested in this issue." Excuse me but I don't think so.

(3) There are also other similiar sentences that need to rephrease.

In addition, it is not appropriate to examine both adaptation and population structure using this one gene, as neutral sites are used for population sturcture whereas selected sites are used for adaptation. Rephrease "population structure of A. cerana" to "structure of Vg gene" etc. Although it is possible to use markers from Vg gene to identify certain populations, it does not represent population structure.

Other typo/mistakes:

line 330, should be "dN/dS < 1

line 91: "-800 degrees"

line 159 says "from 0.236 to 0.999" while line 204 says "from 236 to 999", the latter is unrealistically high. Same for many other values in that paragraph and table 2.

Author Response

Reply to the Reviewer 2

The authors examined the exons of honeybee Vg gene. This gene is very important in honey bees and showed quite some diversity. This study examines the variation of Vg genes in A. cerana populations, with an emphesis on those from northern distribution of A. cerana.
-Thanks to the second reviewer for carefully reading our manuscript. Thank you very much for helping in improving our manuscript. We appreciate your work. All corrections are green highlighted.

I do have some concers over some of the expressions in the manuscript where at least overstate of importance is found:
-we improved the manuscript. All changes are green highlighted.

(1) line 61-64: the phrase "A. c. koreana is the most interesting ...", "most of the honeybee migrations take place through the Korean peninsula"
-these phrases are corrected.

(2) line 250: "all beekeepers and scientists are interested in this issue." Excuse me but I don't think so.
-these phrases are corrected.

(3) There are also other similiar sentences that need to rephrease.
-similiar sentences are rephreased.

In addition, it is not appropriate to examine both adaptation and population structure using this one gene, as neutral sites are used for population sturcture whereas selected sites are used for adaptation. Rephrease "population structure of A. cerana" to "structure of Vg gene" etc. Although it is possible to use markers from Vg gene to identify certain populations, it does not represent population structure.
-the sentence is corrected and rephrased.

Other typo/mistakes:

line 330, should be "dN/dS < 1
-the mistake is corrected.

line 91: "-800 degrees"
-the mistake is corrected.

line 159 says "from 0.236 to 0.999" while line 204 says "from 236 to 999", the latter is unrealistically high. Same for many other values in that paragraph and table 2.
-the mistakes are corrected.